# Integrated Analysis of Gut Microbiome and Liver Metabolome to Evaluate the Effects of Fecal Microbiota Transplantation on Lipopolysaccharide/D-galactosamine-Induced Acute Liver Injury in Mice

**DOI:** 10.3390/nu15051149

**Published:** 2023-02-24

**Authors:** Chunchun Yuan, Jinghui Fan, Lai Jiang, Wenxin Ye, Zhuo Chen, Wenzi Wu, Qixin Huang, Lichun Qian

**Affiliations:** 1Key Laboratory of Animal Nutrition and Feed Science in East China, Ministry of Agriculture, College of Animal Sciences, Zhejiang University, Hangzhou 310058, China; 2Hangzhou Academy of Agricultural Sciences, Hangzhou 310004, China; 3Hainan Institute of Zhejiang University, Sanya 572025, China

**Keywords:** acute liver failure, fecal microbiota transplantation, gut microbiota, liver metabolomics, correlation analysis, hepatic apoptosis

## Abstract

Acute liver failure (ALF) refers to the occurrence of massive hepatocyte necrosis in a short time, with multiple complications, including inflammatory response, hepatic encephalopathy, and multiple organ failure. Additionally, effective therapies for ALF are lacking. There exists a relationship between the human intestinal microbiota and liver, so intestinal microbiota modulation may be a strategy for therapy of hepatic diseases. In previous studies, fecal microbiota transplantation (FMT) from fit donors has been used to modulate intestinal microbiota widely. Here, we established a mouse model of lipopolysaccharide (LPS)/D-galactosamine (D-gal) induced ALF to explore the preventive and therapeutic effects of FMT, and its mechanism of action. We found that FMT decreased hepatic aminotransferase activity and serum total bilirubin levels, and decreased hepatic pro-inflammatory cytokines in LPS/D-gal challenged mice (*p* < 0.05). Moreover, FMT gavage ameliorated LPS/D-gal induced liver apoptosis and markedly reduced cleaved caspase-3 levels, and improved histopathological features of the liver. FMT gavage also restored LPS/D-gal-evoked gut microbiota dysbiosis by modifying the colonic microbial composition, improving the abundance of *unclassified_o_Bacteroidales* (*p* < 0.001), *norank_f_Muribaculaceae* (*p* < 0.001), and *Prevotellaceae_UCG-001* (*p* < 0.001), while reducing that of *Lactobacillus* (*p* < 0.05) and *unclassified_f_Lachnospiraceae* (*p* < 0.05). Metabolomics analysis revealed that FMT significantly altered LPS/D-gal induced disordered liver metabolites. Pearson’s correlation revealed strong correlations between microbiota composition and liver metabolites. Our findings suggest that FMT ameliorate ALF by modulating gut microbiota and liver metabolism, and can used as a potential preventive and therapeutic strategy for ALF.

## 1. Introduction

Acute liver failure (ALF) leads to hepatocytes apoptotic, large amounts inflammatory response, and liver damage, thereby leading to multiple organ failure [1,2]. Numerous factors can stimulate acute liver injury, including hepatitis viruses, drugs, and toxins [3]. At present, the effective therapeutic strategies for ALF are still lacking [4], of which liver transplantation is still the most effective therapy for ALF. There is an urgent need for new therapies and drugs due to the lack of liver donors and extremely high costs [5]. Lipopolysaccharide (LPS), a characteristic component of Gram-negative bacteria, stimulates Kupffer cells, leading to the release of numerous pro-inflammatory cytokines including interleukin-1β (IL-1β), interleukin-6 (IL-6), and tumor necrosis factor-α (TNF-α) [6,7]. D-galactosamine (D-gal) is a hepatocyte disrupting agent, which used to enhance the liver toxicity of LPS [8]. Therefore, the LPS/D-gal-challenged animal model has been well established to investigate the mechanisms and potential therapeutic strategies of ALF [8,9].

Human gut microbiota composition is linked to various diseases, including respiratory, neurological, hepatic, gastroenterological, and cardiovascular disorders [10]. Consequently, many studies have suggested that changes in gut microbiome affect the disease development. As the first organ to come into contact with microbial products that enter the portal circulation across the intestinal epithelial, the liver is likely influenced by microbiome and its metabolites [10,11]. Various acute and chronic hepatic diseases affect the gut microbiota composition, which has impact on the pathogenesis of liver diseases, the pathophysiology of this bidirectional relationship has recently been the subject of several studies [5,12,13]. The use of bacteria as probiotics has a long history. Fecal microbiota transplantation (FMT) refers to transfer intestinal microbes from a fit donor and possible implantation into the intestinal tract of recipients to rebuilt a healthy intestinal microbial ecosystem [14,15]. FMT has been ratified as standard treatment for *Clostridia difficile* infections, and has provided evidence for the effect of the microbiota in various diseases and has become a study hotspot in clinical medicine and biomedicine recently [16,17]. FMT also received interest for its therapeutic potential in autoimmune, cardiac, and other extraintestinal diseases [18,19], including as a viable and safe treatment for a variety of liver diseases. Studies by Schneider KM et al. showed that intestinal microbiome is a potentially modifiable risk factor for ALF, and that ALF can change the composition of intestinal microbiome, leading to intestinal barrier damage and bacterial translocation [20]. Ferrere et al. found that fecal transplantation from alcohol-resistant donor mice prevented dysbiosis and alcohol-caused liver injury [21]. Wang et al. revealed that FMT prevented hepatic encephalopathy in rats with acute hepatic dysfunction evoked by carbon tetrachloride [18]. The function of FMT in the therapy of chronic hepatic disease and delaying the progression of liver cancer provides new horizons for the clinical therapy of hepatic disease [22]. FMT also has significant advantages in reducing medical and social costs.

Therefore, we hypothesized that there are probiotics against liver injury in the gut microbes of animals, and that FMT can effectively treat ALF. We assessed the preventive and therapeutic functions of FMT from healthy mice on LPS/D-gal caused ALF in C57BL/6 mice. Additionally, 16S rDNA and metabolomics technology were used to analyze FMT-mediated changes in gut microbiota and liver metabolome, and the potential mechanisms of hepatoprotective effects of FMT.

## 2. Materials and Methods

### 2.1. Animals

Male C57BL/6 mice (SPF-grade, Hans Biotechnology Co., Ltd., Hangzhou, China), 6 weeks old, were provided with fixed temperature (21–24 °C), humidity (50–60%), and natural lighting, and given water ad libitum. All experiment were carried with the guidelines for animal care and use, and were approved by the Committee for Animal Research at Zhejiang University (Hangzhou, China) (ZJU20220438).

### 2.2. Experimental Protocol

Mice were intraperitoneally injected with LPS and D-gal, purchased from Sigma-Aldrich, to induce ALF. After 1 week of acclimation, the animals were separated into six groups randomly with 10 mice in each group: (1) control, (2) LPS/D-gal, (3) LPS/D-gal combined with FMT preventive treatment for 7 days (FMT-pre7d group), (4) LPS/D-gal + FMT preventive treatment for 14 days (FMT-pre14d group), (5) LPS/D-gal + FMT treatment for 7 days (FMT-treat7d group), and (6) LPS/D-gal + FMT treatment for 14 days (FMT-treat14d group). Prevention groups were gavaged with FMT (0.2 mL, 1 × 10^9^ CFU/mL) once a day for 7 or 14 days before LPS/D-gal administration. Additionally, the treatment groups were gavaged with FMT (0.2 mL, 1 × 10^9^ CFU/mL) once a day for 7 or 14 days after LPS/D-gal administration. The preventive and therapeutic effects of FMT were assessed over two time periods to determine its effect on acute liver injury over time. Correspondingly, the control and LPS/D-gal model groups were gavaged 0.2 mL PBS once a day. After 14 days, the preventive and therapeutic groups were injected with D-gal and LPS intraperitoneally at the doses of 400 mg/kg and 50 μg/kg body weight, respectively. Meanwhile, the control group was injected the same dose of PBS.

On day 14, the mice were anesthetized and sacrificed 6 h after injection. Professors with Medical Laboratory Animal and animal Experiment Certificate of Zhejiang Province handled the experimental animals. The serum, colonic content, and liver tissues were collected immediately from all mice. A part of the liver was fixed using 4% paraformaldehyde solution, and the rest of liver tissue was saved at −80 °C.

Feces from control mice were collected daily as donors. Then, we placed the collected fresh fecal material in sterile tubes, and homogenized in sterile normal saline. Centrifuged at 6000× *g* for 10 min, pass the homogenate through 70 μm filter, and quantified (OD 600 value of 1.0, predicted to be 10^9^ CFU/mL). Eventually, the fecal bacterial slurry was diluted to 5.0 × 10^9^ CFU/mL using 30% sterile medical glycerin PBS, and stored at −80 °C [23]. 

### 2.3. Determination of Liver Enzymes and Biochemical Indexes

A ratio of 1:9 was used for homogenizing 10% liver tissue homogenates, which was then centrifuged at 3000 r/min (4 °C) for 15 min. The activities of alanine aminotransferase (GPT/ALT), aspartate aminotransferase (GOT/AST), and total bilirubin (TBIL) in mice livers were determined according to the instructions (Jiancheng Institute of Biological Engineering, Nanjing, China). Concentrations of cytokines, namely, serum TNF-α, IL-1β, IL-6, IL-10, and IL-12, using mouse ELISA kits (Jiancheng Institute of Bioengineering, Nanjing, China) to assay. 

### 2.4. Terminal Deoxynucleotidyl Transferase-Mediated Nucleotide Nick-End Labeling (TUNEL) Assay

TUNEL assays detected hepatocyte apoptosis by terminal deoxynucleotidyl transferase-mediated dUTP nick-end labeling. To label nuclei, liver sections were counterstained with 4′-6-diamidino-2-phenylindole after TUNNEL labeling. Using a fluorescence microscope to acquired images (Olympus, Shanghai, China).

### 2.5. Histology and Immunohistochemistry

The livers were fixed using 4% paraformaldehyde, and embedded using paraffin, and stained using hematoxylin and eosin (H and E) or myeloperoxidase (MPO) antibodies. Histopathological changes in each mouse were examined using an optical microscope (Nikon, Tyoko, Japan).

### 2.6. Microbial 16S rRNA Gene Sequencing Analysis

Colon contents were taken on day 14 for microbiota analysis. Extracted fecal DNA from the microbial community using DNA Kit (Omega Biotek, Norcross, GA, USA). The, we utilized 338F and 806R universal PCR amplification to change the regions of 16S rRNA genes V3-V4. The activation was 2 min at 95 °C, denaturation of 30 s at 95 °C, annealing of 30 s at 50 °C, and extension of 30 s at 72 °C, then extension of 10 min at 72 °C followed by 27 cycles. PCR amplicon products were isolated using a 2% agarose gel and a DNA gel extraction kit to purify. We used the QuantiFluor™-ST system (Promega, Madison, WI, USA) to quantify the PCR products. Sequencing was performed on an Illumina MiSeq platform (Illumina, San Diego, CA, USA) at Majorbio, Shanghai, China.

The following criteria were used for demultiplexing and filtering raw reads by QIIME (version 1.9.1): (1) At each site with a mean quality score < 20, 300 bp reads were trimmed, and reads less than 50 bp were deleted after a sliding window of 10 bp; (2) Reject sequences with indistinct reads or two nucleotide mismatches in primer mates. Sequences with at least 97% of matching nucleotides are clustered in the operational taxonomic units (OaTUs). Every 16S rRNA gene sequence was explained using the RDP classifier (version 2.11), compared to the SILVA 16S rRNA database (version 132), and set a comparison threshold of 70% [24,25].

### 2.7. Metabonomics

Non-targeted profiling of metabolites in mouse liver tissues were performed on the Majorbio Bio-Pharm Technology Co., Ltd. (Shanghai, China). First, liver samples (50 mg) were spiked with 400 μL of cold methanol (LC/MS grade, A456−4) solution (80%, *v*/*v*). Homogenized tissue sample matrix using an automated homogenizer (FastPrep-24TM5G; MP Biomedicals, Santa Ana, CA, USA). Then, the samples were vortexed 30 s and extracted metabolites by sonication for 10 min in an ice water bath. The extraction process was repeated thrice. Then, incubated samples 30 min (−20 °C) and centrifuged 13,000× *g* 15 min at 4 °C (Centrifuge 5424-R, Eppendorf, Hamburg, Germany). Then, collected 200 μL of supernatant for liquid chromatography–mass spectrometry (LC–MS).

LC–MS uses a Waters ACQUITY ultraperformance liquid chromatography system with a triple time-of-flight mass spectrometer (Waters, Milford, CT, USA; AB SCIEX TripleTOF 5600 System, Framingham, MA, USA). Separate chromatographic on Waters ACQUITY UPLC R BEH C18 (100 mm × 2.1 mm, 1.7 μm) at a flow rate of 0.4 mL/min preheated to 40 °C. The mobile phase consists of ultrapure water including 0.1% formic acid (phase A), and a mixture of acetonitrile: isopropanol with 0.1% formic acid according 1:1 volume (phase B), and injection volume is 20 μL. The gradient started with 5% phase B and 95% phase A, increases to 20% phase B in 3.0 min, increases to 95% in 6.0 min, and holds for 4.0 min, respectively. Thereafter, the gradient turned to the original chromatographic status within 0.1 min and hold 2.9 min. The scanning scope was *m*/*z* 50–1000 and the resolution was 30,000 resolutions for the full-scan mode. Using nitrogen as the carrier and a rate of 900 L/h. Quality parameters are set to ion original temperature is 120 °C, desolvation temperature is 500 °C, electrospray capillary voltage is 1.0 kV, collision voltage is 6 eV, and injection voltage is 40 V. We prepared quality control (QC) samples by mixing all sample extraction aliquots to evaluate system stability and were analyzed after each of the five samples throughout the analytical run. Principal component analysis suggested that the QC samples were clustered closely, verifying the good repeatability of UPLC-triple TOF-MS/MS method.

Metabolomics raw data from LC–MS were analyzed, and calibration was done using the Progenesis QI software (Waters, Milford, USA). Peak intensities were normalized to exclude peaks with the relative standard deviation > 30% in QC samples. Comparing its mass spectrometry information to public and commercial databases (http://www.hmdb.ca/; https://metlin.scripps.edu/; https://i-sanger.com) (accessed on 23 August 2021), annotating metabolites. The data from the positive and negative ion modes were then integrated into the SIMCA-P 14.0 (Umetrics, Umeå, Sweden) for further exploration. Principle component analysis (PCA) using an unsupervised method was applied to obtain an overview of the metabolic data, general clustering, trends, or outliers were visualized. Overall differences between treatments were distinguished using orthogonal partial least squares discriminant analysis (OPLS-DA) and Student’s t-test. The changed metabolites among treatments were analyzed using the importance in the projection (VIP) values > 1.0 and *p*-value < 0.05. To evaluate the potentially affected metabolic pathways, metabolic pathways enrichment, and topological analysis were conducted on the Kyoto Encyclopedia of Genes and Genomes (KEGG) library [26,27].

### 2.8. Western Blot Analysis

Separate proteins electrophoretically on 10% SDS-polyacrylamide gels and the bands were transferred on polyvinylidene difluoride membranes. We blocked the membrane for 1 h using 5% defatted skim milk and incubated with Tris-buffered saline containing anti-caspase-3 at 1:1000 dilution overnight, then incubated with secondary antibody. Finally, we used the ChemiDoc MP (Bio-Rad, Hercules, CA, USA) imaging system to obtain images [28].

### 2.9. Statistical Analysis

Statistical calculations using the GraphPad Prism (version 8.0, San Diego, CA, USA), and expressed as mean ± SEM. We used one-way ANOVA and Tukey’s multiple comparison test to analyze. We used Spearman’s rank correlation test to analyze the correlation of variants. The statistical significance was set at *p* < 0.05.

## 3. Results

### 3.1. FMT Relieved LPS/D-Gal-Induced Liver Injury in Mice

To evaluate liver injury, several hepatic enzymes in liver homogenates were examined. Hepatic plasma ALT and AST, indicators of ALF [29], were remarkably improved in the LPS/D-gal group. Interestingly, FMT prevention and treatment decreased the elevated ALT and AST levels significantly (Figure 1A,B). Furthermore, total bilirubin (TBIL) level, an indicator of liver function, were decreased significantly in the FMT treatment groups than that in the LPS/D-gal group (Figure 1C). As the H and E staining showed, the LPS/D-gal challenge acute hepatic injury accompanied by a range of pathological manifestations, such as hemorrhagic necrosis, hepatic structural destruction, and numerous inflammatory cell infiltration. However, these morphological changes improved in the livers of the FMT prevention and treatment group mice (Figure 1D). In conclusion, these results indicate that pretreatment and treatment with FMT ameliorated ALF in LPS/D-gal-caused mice. Moreover, we also found that the effect of the 14 d groups was better than that of the 7 d groups, and the 14 day treatment group had the best results.

### 3.2. FMT Ameliorated Inflammatory Response in LPS/D-Gal-Challenged Mice

The hepatocellular injury caused by LPS/D-gal is linked with liver and systemic inflammation [30]. Serum IL-6, IL-1β, and TNF-α levels were improved significantly in the LPS/D-gal group, and pretreatment and treatment with FMT effectively suppressed these pro-inflammatory cytokines levels, which further decreased with prolongation of FMT gavage (Figure 2A–C). Moreover, the anti-inflammatory cytokines IL-10 and IL-22 decreased significantly in the LPS/D-gal group and recovered in FMT prevention and treatment groups (Figure 2D,E). Thus, FMT by gavage suppressed LPS/D-gal caused systemic and liver pro-inflammatory responses in mice. A notable increase in MPO, a marker of neutrophil infiltration, was found in the LPS/D-gal group, and treatment with FMT repressed neutrophil infiltration caused by LPS/D-gal (Figure 2F).

### 3.3. FMT Inhibited Apoptosis of Hepatocytes in LPS/D-Gal-Induced Mice

LPS/D-gal caused by ALF is characterized by excessive apoptosis of hepatocytes [31]. Hence, hepatocyte apoptosis was detected using TUNEL staining and caspase activation in the liver. As shown in Figure 3A, numerous TUNEL-positive hepatocytes were observed in LPS/D-gal-treated liver tissues. However, the improved number of positive hepatocytes was lower in the FMT prevention and treatment groups. Similarly, the level of active caspase-3 was upregulated in LPS/D-gal-induced mice, whereas this induction was inhibited in the FMT prevention and treatment groups (Figure 3B). Therefore, FMT by gavage suppressed LPS/D-gal-treated hepatocyte apoptosis and protected against ALF. Moreover, the FMT treatment 14 d group showed a lower positive TUNEL than the FMT treatment 7 d group, suggesting that prolonged FMT administration have better therapeutic effects against liver injury.

### 3.4. FMT Modulated Gut Microbiota Composition in LPS/D-Gal-Induced Mice

We further analyzed the influence of FMT therapy on the colonic microbial composition of LPS/D-gal challenged mice by 16S rDNA gene sequencing. We found decreased α-diversity indices, including Shannon, Sobs, Chao, and ACE indices of the gut microbiota of LPS/D-gal group than those of control group (*p* < 0.05). Aforementioned indices were increased in FMT prevention and treatment groups than in the LPS/D-gal group, although not obviously (Figure 4A–D), suggesting that FMT increased bacterial richness (as determined by rising Sobs, Chao, ACE, and Shannon indices) and evenness (as determined by rising Shannon index) in LPS/D-gal-induced mice. Then, we analyzed the β-diversity with the Bray–Curtis principal coordinate analysis (PCoA). PCoA exposed remarkable differences in gut microbiota between the LPS/D-gal and control groups (R^2^ = 0.4343, *p* = 0.002, Figure 4E). The LPS/D-gal and FMT treatment groups also showed different gut microbiota (R^2^ = 0.3216, *p* = 0.001, Figure 4G), but there was no significant difference between LPS/D-gal and FMT prevention groups (R^2^ = 0.2974, *p* = 0.001, Figure 4F). These results indicated that FMT modulates gut microbiota composition of the ALF mice are induced by LPS/D-gal.

Subsequently, we assessed the relative abundance of bacteria on phylum and genus levels (Figure 4H,I). *Bacteroides* and *Firmicutes* were the most abundant, accounting for over 80% of all microorganisms in all groups. In Figure 4J, the relative abundance of *Bacteroidota* was reduced (*p* < 0.01), and those of *Firmicutes* (*p* < 0.01) and *Desulfobacterota* (*p* < 0.01) were increased in LPS/D-gal-challenged mice. FMT treatment inverted this change by improving the relative abundance of *Bacteroidetes* (*p* < 0.001) and reducing the abundance of *Firmicutes* (*p* < 0.01) and *Desulfobacterota* (*p* < 0.05). Compared with the FMT treatment groups, FMT prevention groups mainly showed an increased abundance of *Bacteroidetes* (*p* < 0.001) and a reduced abundance of *Firmicutes* (*p* < 0.01). Furthermore, within the phylum *Bacteroidota* (Figure 4K), the abundance of *norank_f_Muribaculaceae* (*p* < 0.01), *Alistipes* (*p* < 0.05), and *Prevotellaceae_UCG-001* (*p* < 0.01) was significantly decreased in LPS/D-gal than control group and accounted for the majority of reduced abundance of *Bacteroidetes* in LPS/D-gal group. Improved abundance of *Firmicutes* in LPS/D-ga-induced mice was mostly explained by the improved abundance of *Lactobacillus* (*p* < 0.05) and *unclassified_f_Lachnospiraceae*.

As shown in Figure 4L, FMT prevention and treatment mitigated LPS/D-gal-induced changes in bacterial abundance. FMT prevention improved the abundance of *norank_f_Muribaculaceae* (*p* < 0.01) and reduced that of *Lactobacillus* (*p* < 0.05). FMT treatment improved the abundance of *norank_f_Muribaculaceae* (*p* < 0.001), *Alloprevotella* (*p* < 0.05), *unclassified_o_Bacteroidales* (*p* < 0.001), and *Prevotellaceae_UCG-001* (*p* < 0.001), and reduced that of *unclassified_f_Lachnospiraceae* (*p* < 0.05) (Figure 4M). Therefore, FMT treatment alleviated gut dysbiosis by modulating gut microbiota composition in LPS/D-gal challenged mice. 

### 3.5. FMT Altered Liver Metabolome in LPS/D-Gal-Induced Mice

Analysis of the liver metabolome using LC-MS-based non-targeted metabolomics. PCA showed a distinct variation in liver metabolome of the control and LPS/D-gal groups (Figure 5A,B), suggesting that hepatic injury induced metabolic disturbances. The liver metabolome of FMT prevention and treatment group mice were slightly overlapped with those of control group and were distinct from that of LPS/D-gal group, clearly indicating the role of FMT in preventing and treating hepatic injury. Furthermore, PLS-DA exhibited a better discriminative ability than PCA and was used to analyze the metabolic profiles. The OPLS-DA score plot showed significant distinctions in the liver metabolite content of the six groups (Figure 5C,D). The hierarchical clustering heatmap in Figure 5E shows the changing trends of liver metabolites, and indicating significant differences in the control, LPS/D-gal, and FMT prevention and treatment groups.

Based on variable VIP > 1, fold change (FC) ≥ 1.2 or ≤0.8333, *p* < 0.05, and with KEGG annotations. As shown in Table 1, compared with the control group, there were 25 significantly differentially expressed metabolites in the LPS/D-gal group; of which, 11 were upregulated and 14 were downregulated—vs. (Table 1). The levels of these metabolites tended to restore in the FMT prevention and treatment groups. We found that caffeine, L-2-Aminoadipic acid, cortolone, thymine, and 12(R)-HETE were significantly downregulated in the FMT-pre7d group, and cortolone, N-acetyl-alpha-D-glucosamine 1-phosphate, and taurocholate were significantly down-regulated in the FMT-pre14d group. Ne-acetyllysine, trichloroethanol glucuronide, thiamine, 2-(S-glutathionyl)acetyl glutathione, stachyose, and 5-L-Glutamyl-L-alanine were significantly up-regulated in both the FMT-pre7d and 14d groups. These metabolites were further studied. KEGG enrichment analysis revealed that FMT prevention mainly affected lysine degradation, metabolism of xenobiotics by cytochrome P450, the biosynthesis of plant secondary metabolites and other pathways (Figure 5G,H).

Compared with the LPS/D-gal group, both the FMT-treat7d and FMT-treat14d groups exhibited a decrease in metabolites of nucleotide sugar and amino sugar metabolites, such as UDP-N-acetyl-alpha-D-glucosamine, N-acetyl-alpha-d-glucosamine 1-phosphate, and CDP-glucose, and an increase in metabolites issued from xenobiotic metabolism by cytochrome P450 (trichloroethanol glucuronide, 2-(S-Glutathionyl)acetyl glutathione, and 1-methylguanosine), and starch and sucrose metabolites (UDP-glucose, CDP-glucose, and maltose). In addition, we found that cholesterol metabolites (number of metabolites = 2, *p* = 0.0005) were down-regulated in the FMT-treat 7 d group significantly and arachidonic acid metabolites (number of metabolites = 4, *p* = 0.0002) were notably down-regulated in the FMT-treat 14 d group (Figure 5I,J).

### 3.6. Gut Microbiota Is Associated with Liver Metabolites and Inflammatory Markers

The correlations between gut microbes, liver metabolites, and markers of inflammation were analyzed to further explore the potential mechanism of action of FMT. Figure 6A,B displayed the correlation between differential gut microbes on phylum, and genus and inflammatory markers. *Firmicutes* and *Desulfobacterota* showed a positive relationship with AST, ALT, TBIL, IL-6, IL-1β, and TNF-α, and a negative correlation with IL-10 and IL-22 levels. Similar trends were observed at the genus level, including *Lactobacillus*, *Lachnospiraceae_UCG-006 unclassified_f_Lachnospiraceae*, *Desulfovibrio*, etc. Opposite correlations were observed in *Bacteroides*, *unclassified_o__Bacteroidales*, *norank_f__Muribaculaceae*, and *Prevotellaceae_UCG-001*, etc. Figure 6C showed the correlation between metabolic biomarkers and inflammatory markers. Thymine, CDP-glucose, and N-acetyl-alpha-D-glucosamine 1-phosphate were negatively correlated with anti-inflammatory cytokines and direct correlation with pro-inflammatory cytokines. In particular, *unclassified_o__Bacteroidales*, *unclassified_f__Prevotellaceae*, and *norank_f__Muribaculaceae*, among others positively correlated with metabolites, such as trichloroethanol glucuronide, stachyose, and maltose, that were notably downregulated in LPS/D-gal group, and negatively correlated with some metabolites that were significantly upregulated in LPS/D-gal group. Conversely, *Lachnoclostridium*, *Faecalibaculum*, *Dubosiella*, and *Lachnospiraceae_UCG-006* showed opposite trends (Figure 6C).

## 4. Discussion

ALF, fulminant hepatic failure, has a high mortality rate and resource cost, which is caused by the stimulation of drugs and toxins. Clinical manifestations of ALF include rapid hepatic injury, derangements in coagulopathy, hepatic encephalopathy, and multi-organ failure, which pose severe threats to a patient [32,33]. D-Gal, a hexosamine derived from galactose, is a hepatotoxic poison, and intraperitoneal administration of which will induce multiple hepatocyte damage and inflammation, is analogous to the pathological status of clinical viral hepatitis [34,35]. Combined use of LPS and D-gal has been diffusely applied to set up animal models of ALF [36]. Considering the limitations of the existing ALF treatments, there is urgent to find effective treatments to ameliorate ALF-related treatment difficulties. There is increasing evidences that the dynamic changes of intestinal microbiota perform an important role in the occurrence and development of liver injury. In view of this, the regulation of intestinal microbiota are promising diagnostic, prognostic, and therapeutic tools. In this regard, FMT is the most powerful tool for resetting intestinal microbiome disturbances caused by liver diseases. [37,38]. Our study mainly explored the guarding role of FMT and its potential mechanism of action in an LPS/D-gal-induced ALF, and found that FMT can be used as a safe, effective, and low-cost treatment for ALF. However, there is no clear protocol for the standardized donor screening, preparation, pretreatment, administration, or long-term stability of FMT, and the exact mechanism involved in its action has not been elucidated. Furthermore, histopathological assessment and aminotransferase activities revealed that FMT intervention also alleviated LPS/D-gal-challenged hepatic failure. FMT decreased hepatic ALT and AST levels significantly to attenuate liver failure, which was confirmed by the decrease in TUNEL-positive hepatocytes, indicating reduced hepatocyte apoptosis. LPS, on administration, binds TLR4 to activate it, which in turn activates the NF-κB via intermediate proteins, which then induces the expression of inflammatory cytokines [39,40]. We found that FMT restrained LPS/D-gal-caused hepatic neutrophil infiltration, and the making of pro-inflammatory cytokine and chemokine. In ALF, hepatocytes and other cells produce excessive inflammatory mediators, resulting partial and systemic inflammation [41]. Sustained making of cytokines including TNF-α, IL-1β, IL-6, IL-10, and IL-22, causes hepatocellular inflammation and apoptosis, and eventually leads to ALF [42,43]. According to the research, inhibition of inflammatory cytokines making may be a potential tactic for the therapy of ALF [43,44]. We found that FMT intervention downregulated the pro-inflammatory cytokines levels significantly and upregulated the anti-inflammatory cytokines, thereby reducing LPS/D-gal-induced inflammation, consistent with a previous study [45,46].

Studies have shown that the TNF-α is the main pro-apoptotic factor inducing extensive hepatocyte apoptosis in LPS/D-gal caused ALF [47,48]. TNF-α induces the neutrophil migration, that casts an important role in hepatocyte necrosis in the late stage of ALF [49]. TNF-α administration has been improved to accelerate hepatic failure [50], while TNF-α production inhibition [51] or TNF-α knockout [48] efficiently prevents acute liver injury. We found that FMT inhibited TNF-α production, a key regulator of apoptosis, to ameliorate the development of LPS/D-gal-induced ALF. Daubioul et al. [52] demonstrated in an initial research of seven patients with nonalcoholic steatohepatitis (NASH), prebiotic feeding for 8 weeks significantly reduced levels of liver inflammatory markers. Another study found the modulation of gut microbiota significantly decreased inflammatory markers (TNF-α and C reactive protein) [53]. To sum up, these data suggest that gut microbiota composition and metabolic activity may help suppress hepatic and systemic inflammation. The infiltration of neutrophils into liver tissue increases after the development of ALF, which can promote liver injury by producing inflammatory mediators [54]. Therefore, we measured the level of the neutrophil infiltration index MPO. We observed that FMT reduced MPO activity, indicating decreased neutrophil filtration. TNF-α induces apoptosis and necrosis of hepatocytes by activating caspase-3, which is a key executive molecule in apoptotic pathways [55,56]. FMT inhibited the expression of caspase-3, which was raised in LPS/D-gal challenged liver, indicating that FMT may adjust cellular apoptosis of ALF by suppressing caspase-3 expression. Finally, we found that prolonged intragastric administration of FMT moderated ALF caused by LPS/D-gal, and the effect was slightly better in the treatment groups than in the prevention groups.

To probe the mechanism of action of FMT, we evaluated the FMT-mediated changes in intestinal microbiota and liver metabolic pathways in mice. The intestinal microbiota is closely associated with health and numerous host functions, including growth, development, immunity, metabolism, and disease [57,58,59]. The liver secretes bile acids that end up in the small intestine. The portal vein that enters the liver is rich in nutrients, and is the main blood provide to liver. The gut-liver axis makes reference to the interplay between the gut (including the microbes in it) and the liver [11]. Accumulating studies revealed that the composition and metabolites of gut microbes can regulate and promote the development of the human immune system, such as through probiotics, prebiotics, antibiotics, FMT, diet regulation, and management of bacterial complexes [60,61].

FMT can replenish a healthy colonic microbial condition and restore colonization [62]. It is currently an established clinical therapy for *Clostridium difficile* infection [63] and was explored as a therapy for ameliorating inflammatory bowel disease [64], metabolic syndrome [65], and several liver diseases [66]. To elucidate the mechanisms by which FMT manifests its effects, we studied the alterations of colonic microbiota composition and diversity. We found that FMT significantly altered the relative abundance of some phylum and genera in intestinal microbiota, some of which are supported by previous studies. Using 16S rDNA sequencing, it found that LPS/D-gal decreased the abundance of *Bacteroidota* and improved that of *Firmicutes* and *Desulfobacterota* significantly. In contrast, gavage of FMT decreased the *Firmicutes/Bacteroidetes* ratio, which can assess the gut microbial community stability [67]. Therefore, FMT may alleviate liver injury by modulation of gut microbiota composition. Yu et al. [68] also identified an improved abundance in *Firmicutes,* and a reduction in *Bacteroidetes* after *S. boulardii* intervention. *Firmicutes* and *Bacteroidetes* phyla comprise the vast majority of the dominant human gut microbiota [69]. Members of *Bacteroidota* are widely considered as candidate probiotics for the treatment of immune dysfunction, intestinal colitis, metabolic disorders, and cancer [67,70]. The abundance of *norank_f_Muribaculaceae* increased in the FMT prevention groups, while the abundance of *norank_f_Muribaculaceae*, *unclassified_o_Bacteroidales,* and *Prevotellaceae_UCG-001* increased in the FMT treatment groups. *Muribaculaceae* have favorable influences on gut dysregulation through immune modulation and regulation of intestinal homeostasis [71]. The LPS/D-gal-induced reduction in *Muribaculaceae*, which was negatively correlated with AST, ALT, and pro-inflammatory cytokines, was significantly improved after treatment with FMT. Although some strains of the Prevotellaceae appear to be inflammatory pathogens, they act a key role in immunomodulatory, especially those of Prevotella [12]. Thus, the reduced abundance of *Prevotellaceae_UCG-001* in the FMT treatment groups suggested that FMT promoted the abundance of microbes regulating immunity. *Alistipes*, a comparative new genus of *Bacteroidetes*, is highly relevant to immune modulation, dysbiosis, and inflammation [72,73]. Studies have revealed that *Alistipes* abundance is decreased in liver fibrosis diseases, including NASH and nonalcoholic fatty liver disease (NAFLD) [73], as we found. In this research, *Alistipes* likely contributed to worsening liver injury and inversely correlated with IL-1β level. 

The reduction in member of *Bacteroidota* induced by LPS/D-gal was negatively related to pro-inflammatory cytokine levels, which may affect the host immune system. After treatment with FMT, LPS/D-gal caused the depletion of *Muribaculaceae*, *Bacteroidales,* and *Prevotella* in mice to improve. Therefore, FMT may promote immune homeostasis by regulating the intestinal microbiota, which, in turn, modifies the making of pro- and anti-inflammatory cytokines. *Firmicutes* abundance shows positively correlated with pro-inflammatory cytokines and marker enzymes of liver injury. The increase in *Firmicutes* was mainly manifested through the increased abundance of *Lactobacillus* and *unclassified_f_Lachnospiraceae* in the LPS/D-gal group. *Lachnospiraceae* and *Lactobacillaceae* can sugars and hydrolyze starch to generate short-chain fatty acids (SCFAs) [74,75]. Large amounts of SCFAs do not ever confirm favorable effects [76]. The gut microbiota of patients with NAFLD is enriched with *Lachnospiraceae*, as do patients with NASH or obvious hepatic fibrosis [77]. In our research, the gavage treatment of FMT obviously attenuated the abundance of *Lachnospiraceae***,** which was positively correlated with pro-inflammatory cytokines. *Lactobacillus* provides the host with multiple benefits, such as suppression of pathogens and improvement of immune response [78]. Jiang et al. [12] found that pretreatment with probiotic *Lactobacillus reuteri* DSM 17,938 in rats could alleviate intestinal dysbiosis, reduced the inflammatory factors, and alleviated D-gal caused liver injury. However, our results showed a negative impact of *Lactobacillus* for it was positively correlated with pro-inflammatory cytokines, and a significantly negative correlation with IL-10 and IL-22. FMT pretreatment significantly attenuated the abundance of *Lactobacillus,* which is the same as that observed by Yan R et al. [79], who found *Lactobacillus* was heavily enriched in the D-gal-induced group and depleted in Lactobacillus casei strain Shirota (LcS) group.

The metabolomic PLS-DA analysis showed the mice treated with LPS/D-gal had unique metabolic characteristics compared to control group. A total of 25 differentially accumulated metabolites were identified. Through KEGG enrichment analysis, these metabolites were mainly enriched in primary bile acid biosynthesis, bile secretion, cholesterol metabolism, amino sugar and nucleotide sugar metabolism, secondary bile acid biosynthesis, and starch and sucrose metabolism. KEGG analysis revealed that prevention with FMT mainly improved lysine degradation, the metabolism of xenobiotics by cytochrome P450, and the biosynthesis of plant secondary metabolites. The levels of L-2-aminoadipic acid and N-acetyl-alpha-D-glucosamine 1-phosphate, intermediate products of lysine metabolism, were elevated in the LPS/D-gal induced live. Several amino acids are known to have favorable influences on the treatment of liver steatosis, and lysine is known to regulate lipid metabolism in liver [80]. FMT significantly restored the levels of these two metabolites by altering lysine metabolism to alleviate LPS/D-gal-induced liver failure. Recent studies reported the active role of the hexosamine biosynthesis pathway (HBP) in host antiviral immunity of hepatitis B and C virus [81,82]. About 2–5% of the total glucose is transformed to uridine diphosphate N-acetylglucosamine (UDP-GlcNAc)—the last product of HBP [83]. D-gal can exhaust the uridine phosphate in hepatocytes, reducing the production of nucleic acid and protein synthesis [84]. We also found that N-acetyl-alpha-D-glucosamine 1-phosphate, UDP-N-acetyl-alpha-D-glucosamine, and CDP-glucose were markedly downregulated in the FMT treatment groups than LPS/D-gal group; therefore, FMT modulates HBP markedly. Pearson’s correlation analysis revealed that the three metabolites mentioned above positively correlated with pro-inflammatory cytokines. The cytochrome P450 metabolism pathway has important function in detoxification, cell metabolism, and homeostasis of exogenous drugs [85]. Metabolomics results showed that the metabolites of xenobiotics metabolism by cytochrome P450, including 2-(S-Glutathionyl) acetyl glutathione and trichloroethanol glucuronide, which were negatively correlated with pro-inflammatory cytokines, were downregulated remarkably in LPS/D-gal group, and were restored in FMT prevention and treatment groups. Several studies have revealed that CYP450 enzyme are essential in the detoxification of Aristolochic acid I (AAI)-induced liver injury in mice (87). We also found that arachidonic acid metabolism was obviously down-regulated in FMT-treat14d group. As a plentiful lipid mediator in the human, arachidonic acid is essential in the inflammatory metabolic network, and its metabolites have pro-inflammatory features [86,87]. CYP450 hydrolyzes arachidonic acid, and the main metabolite of which is important in pro-inflammatory reactions [88,89]. Correlation analysis revealed arachidonic acid metabolites were positively related to IL-1β, IL-6, and TNF-α levels. 

## 5. Conclusions

In summary, this study demonstrated the preventive and therapeutic effects of FMT against ALF caused by LPS/D-gal; indeed, we found that with prolongation of intragastric administration of FMT, liver failure caused by LPS/D-gal had markedly improved, and the results of treatment were significantly better than those of prevention. Furthermore, changes in the gut microbiome and hepatic metabolism have also been shown to contribute to understanding of the potential mechanism of FMT therapy for ALF. In conclusion, this study suggests that FMT could be a potential therapeutic approach to treat ALF and that the gut microbiome may be a potential therapeutic target for ALF.

## Figures and Tables

**Figure 1 nutrients-15-01149-f001:**
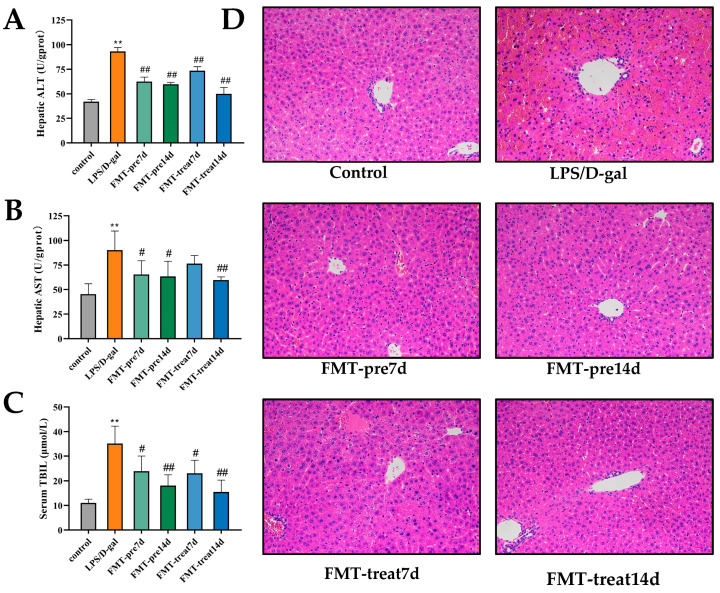
FMT relieved LPS/D-gal-induced acute liver failure (ALF) in mice. (**A**,**B**) Hepatic levels of alanine aminotransferase (ALT) and aspartate aminotransferase (AST) (*n* = 6 in each group). (**C**) Serum level of total bilirubin (TBIL) (*n* = 6). (**D**) Representative images of H and E staining of liver sections (magnification ×200). ** *p* < 0.01 vs. control group, # *p* < 0.05, ## *p* < 0.01 vs. LPS/D-gal group. Dates were expressed as means ± SEM.

**Figure 2 nutrients-15-01149-f002:**
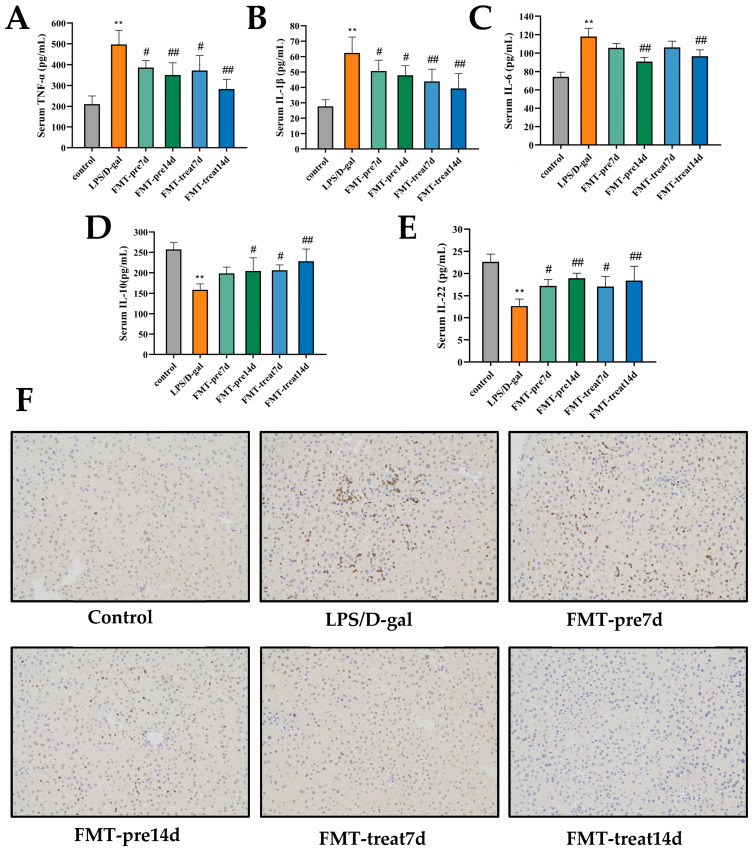
FMT ameliorated inflammatory response in LPS/D-gal-induced mice. (**A**–**E**) Serum TNFα, IL-1β, IL-6, IL-10, and IL-22 levels (*n* = 6). (**F**) Micrographs of liver sections stained with MPO antibody (magnification ×200). ** *p* < 0.01 vs. control group, # *p* < 0.05, ## *p* < 0.01 vs. LPS/D-gal group. Dates were expressed as means ± SEM.

**Figure 3 nutrients-15-01149-f003:**
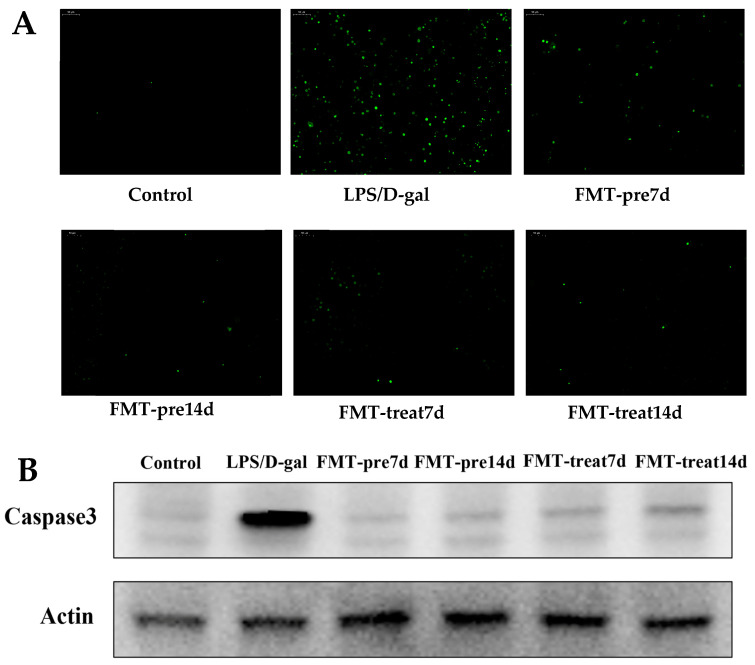
FMT suppressed apoptosis of hepatocytes in LPS/D-gal-induced mice. (**A**) Representative images of hepatic TUNEL staining. Green staining represented apoptotic cells (magnification ×20). (**B**) The expression of Caspase-3 was detected by WB.

**Figure 4 nutrients-15-01149-f004:**
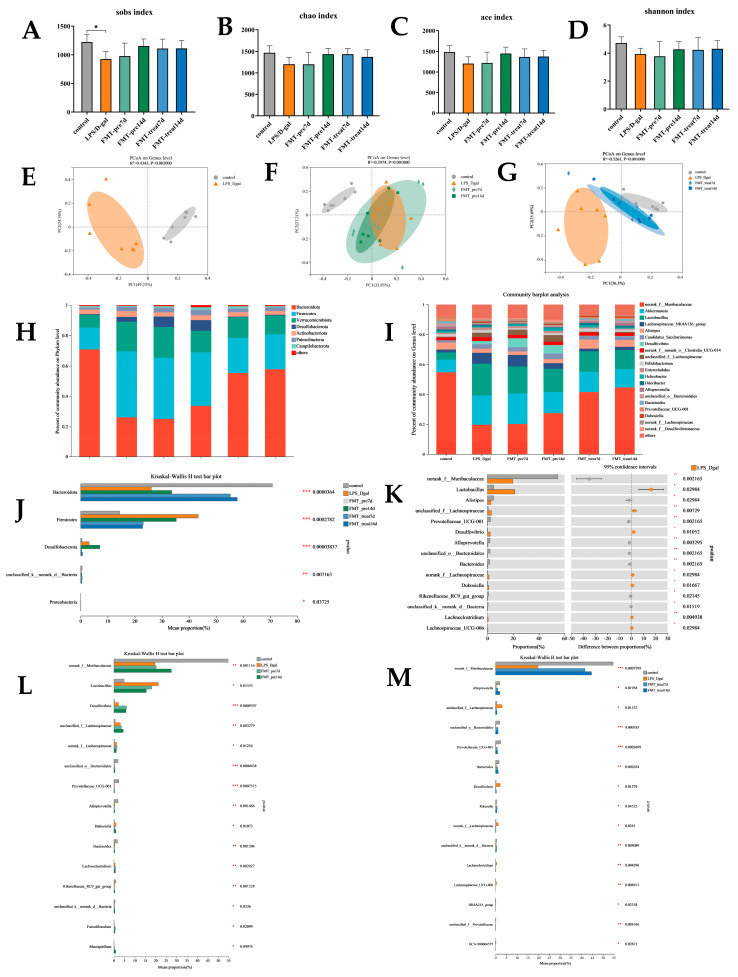
FMT modulated gut microbiota composition. (**A**–**D**) Sobs, Chao, Ace, and Shannon index of OUT level. (**E**–**G**) Weighted UniFrac based principal coordinates analysis (PCoA) analysis. (**H**,**I**) The relative abundance of bacteria on phylum and genus. (**J**) Comparative analysis of the relative abundance of bacteria at phylum level. (**K**–**M**) Comparative analysis of the relative abundance of bacteria on genus level. (*n* = 6–7). * *p* < 0.05, ** *p* < 0.01, *** *p* < 0.001. Data were expressed as means ± SEM.

**Figure 5 nutrients-15-01149-f005:**
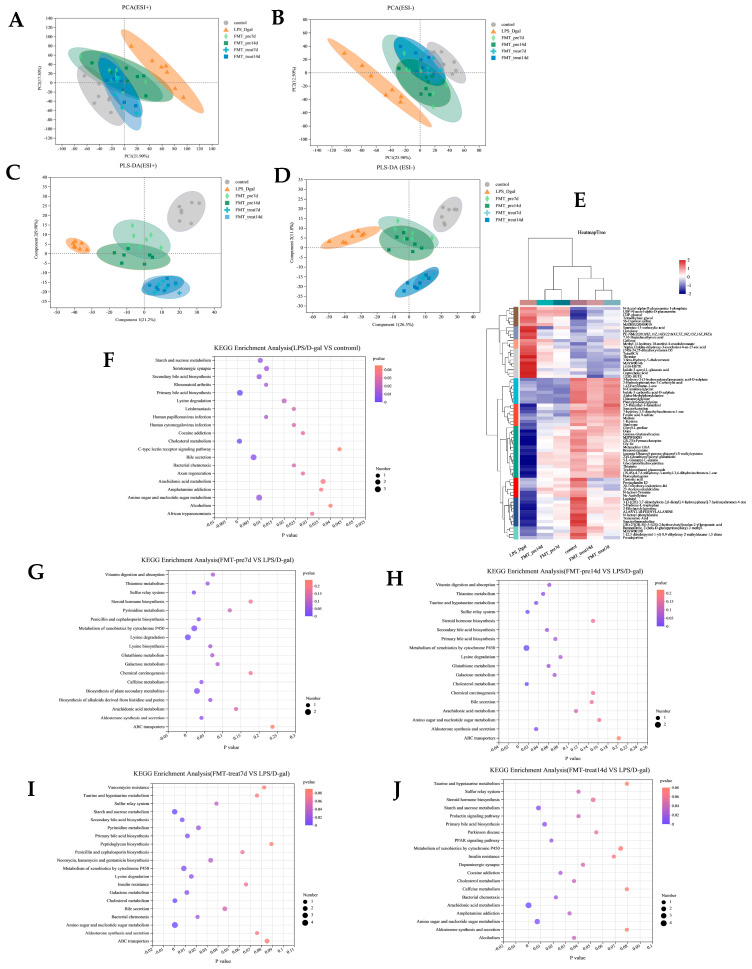
FMT altered liver metabolome. (**A**,**B**) PCA score plots f nontargeted metabolite profiling of the liver samples among the six groups in both positive (ESI+) and negative ionization modes (ESI−). (**C**,**D**) PLS-DA scores of metabolite analysis in ESI+ and ESI− modes. (**E**) Heatmap of liver metabolites from six groups. (**F**–**J**) Bubble diagram showing the KEGG enrichment analysis. Shade of color represents the differences, while bubble size represents the number of enrichments. (*n* = 6–7).

**Figure 6 nutrients-15-01149-f006:**
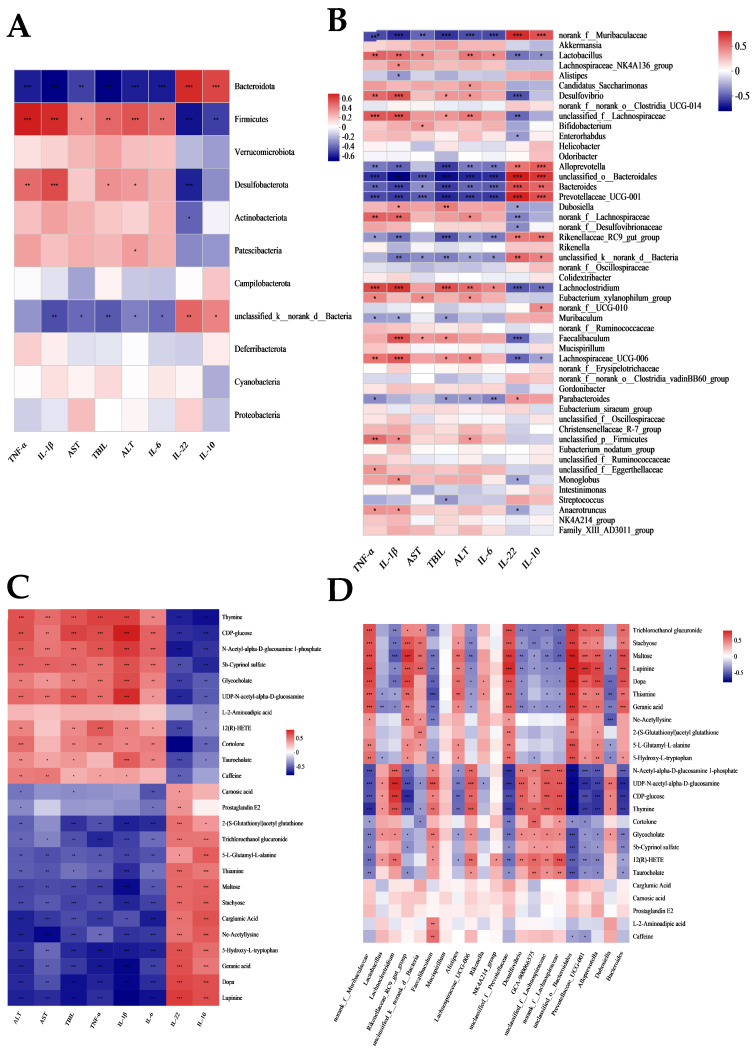
Gut microbiota is linked with liver metabolites and inflammatory markers. (**A**,**B**) Correlations of altered gut microbes with biochemical Indicators on phylum and genus. (**C**) Correlations of altered metabolites with biochemical Indicators. (**D**) Correlation between differential microbiota and metabolic biomarkers. * *p* ≤ 0.05, ** *p* ≤ 0.01, *** *p* ≤ 0.001.

**Table 1 nutrients-15-01149-t001:** Twenty-five metabolites with statistical differences identified based on VIP > 1 in the loading plot, FC ≤ 1.2 or FC ≤ 0.8333, *p* < 0.05, and with KEGG numbers.

						LPS/D-Gal vs. Control		FMT-Pre7d vs. LPS/D-Gal		FMT-Pre14d vs. LPS/D-Gal		FMT-Treat7d vs. LPS/D-Gal		FMT-Treat14d vs. LPS/D-Gal	
No	Metabolite	Formula	Subclass	Mass (M/Z)	RT (min) ^1^	FC ^2^	*p* Value	FC	*p* Value	FC	*p* Value	FC	*p* Value	FC	*p* Value
1	2-(S-Glutathionyl)acetyl glutathione	C_22_H_34_N_6_O_13_S_2_	Amino acids, peptides, and analogues	675.1366	2.7876	0.7921	0.0002	1.2361	0.0006	1.2411	0.0005	1.2571	0.0006	1.2290	0.0007
2	Stachyose	C_24_H_42_O_21_	Carbohydrates and carbohydrate conjugates	701.1902	0.9816	0.6992	0.0000	1.2223	0.0233	1.2190	0.0290	1.3679	0.0008	1.3206	0.0010
3	12(R)-HETE	C_20_H_32_O_3_	Others	319.2273	6.7749	1.2619	0.0000	0.8245	0.0001	0.8154	0.0000	0.7490	0.0000	0.6675	0.0000
4	Carnosic acid	C_20_H_28_O_4_	Diterpenoids	331.1909	6.4796	0.8202	0.0032	/ ^3^	/	/	/	/	/	/	/
5	Trichloroethanol glucuronide	C_8_H_11_C_l3_O_7_	Carbohydrates and carbohydrate conjugates	646.9061	4.3392	0.7386	0.0010	1.2427	0.0156	1.2537	0.0090	1.3571	0.0019	1.2849	0.0041
6	Maltose	C_12_H_22_O_11_	Carbohydrates and carbohydrate conjugates	387.1138	0.9994	0.7691	0.0110	/	/	/	/	1.2519	0.0394	1.2345	0.0373
7	UDP-N-acetyl-alpha- D-glucosamine	C_17_H_27_N_3_O_17_P_2_	Others	628.0553	1.0787	2.5670	0.0000	/	/	/	/	0.7503	0.0292	0.5881	0.0001
8	L-2-Aminoadipic acid	C_6_H_11_NO_4_	Amino acids, peptides, and analogues	160.0605	0.9994	1.2102	0.0029	0.7811	0.0006	/	/	0.8275	0.0090	1.3381	0.0106
9	Ne-Acetyllysine	C_8_H_16_N_2_O_3_	Indolyl carboxylic acids and derivatives	187.1079	2.6841	0.6637	0.0002	1.4079	0.0015	1.3163	0.0146	1.2880	0.0363	/	/
10	Lupinine	C_10_H_19_NO	Others	214.1442	6.5576	0.7915	0.0000	/	/	/	/	/	/	/	/
11	Prostaglandin E2	C_20_H_32_O_5_	Eicosanoids	373.2008	6.6143	0.7992	0.0297	/	/	/	/	/	/	/	/
12	Cortolone	C_21_H_3_4O_5_	Hydroxysteroids	387.2148	6.7749	1.4877	0.0000	/	/	/	/	0.7391	0.0000	0.7845	0.0000
13	Glycocholate	C_26_H_43_NO_6_	Amino acids, peptides, and analogues	464.3007	6.4862	1.2422	0.0000	/	/	/	/	0.8218	0.0000	/	/
14	Geranic acid	C_10_H_16_O_2_	Monoterpenoids	213.1125	5.3076	0.8306	0.0000	/	/	/	/	/	/	/	/
15	5-Hydroxy-L-tryptophan	C_11_H_12_N_2_O_3_	Tryptamines and derivatives	219.0769	2.9159	0.7723	0.0000	/	/	/	/	/	/	/	/
16	5-L-Glutamyl-L-alanine	C_8_H_14_N_2_O_5_	Amino acids, peptides, and analogues	217.0823	1.4357	0.7984	0.0000	1.2053	0.0000	1.2088	0.0000	1.2413	0.0000	/	/
17	CDP-glucose	C_15_H_25_N_3_O_16_P_2_	Pyrimidine nucleotide sugars	564.0628	0.9816	1.9616	0.0000	/	/	/	/	0.7848	0.0000	0.7113	0.0000
18	Carglumic Acid	C_6_H_10_N_2_O_5_	Amino acids, peptides, and analogues	189.0508	0.9504	0.8204	0.0000	/	/	/	/	/	/	/	/
19	Taurocholate	C_26_H_45_NO_7_S	Bile acids, alcohols and derivatives	480.2791	8.8787	1.2538	0.0011	/	/	0.8226	0.0000	0.7505	0.0002	0.7524	0.0016
20	Dopa	C_9_H_11_NO_4_	Amino acids, peptides, and analogues	239.1030	2.8047	0.8176	0.0000	/	/	/	/	/	/	1.2127	0.0000
21	5b-Cyprinol sulfate	C_27_H_48_O_8_S	Bile acids, alcohols and derivatives	550.3423	6.5822	1.3112	0.0001	/	/	/	/	/	/	0.8069	0.0062
22	Thiamine	C_12_H_16_N_4_OS	Pyrimidines and pyrimidine derivatives	265.1122	0.9118	0.7550	0.0001	1.2805	0.0004	1.2224	0.0066	1.3269	0.0003	1.2948	0.0003
23	Caffeine	C_8_H_10_N_4_O_2_	Purines and purine derivatives	195.0882	3.6826	1.6653	0.0002	0.5319	0.0011	/	/	/	/	0.5070	0.0021
24	Thymine	C_5_H_6_N_2_O_2_	Pyrimidines and pyrimidine derivatives	127.0507	2.8586	1.4026	0.0000	0.8138	0.0000	/	/	0.7455	0.0000	0.7439	0.0000
25	N-Acetyl-alpha-D-glucosamine 1-phosphate	C_8_H_16_NO_9_P	Carbohydrates and carbohydrate conjugates	324.0466	0.9748	4.1378	0.0000	/	/	0.8112	0.0153	0.4614	0.0096	0.4818	0.0036

^1^ RT means retention time. ^2^ FC is the fold change of the metabolite expression between groups. ^3^ No statistical difference between two groups.

## Data Availability

The original Sequencing data exists: https://dataview.ncbi.nlm.nih.gov/object/PRJNA866677 (accessed on 6 August 2022), and Metabonomics date were deposited in www.ebi.ac.uk/metabolights/MTBLS5622 (accessed on 23 August 2022).

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
