# Peer review of "Integrated Analysis of Gut Microbiome and Liver Metabolome to Evaluate the Effects of Fecal Microbiota Transplantation on Lipopolysaccharide/D-galactosamine-Induced Acute Liver Injury in Mice"

_nutrients, 2023, doi:10.3390/nu15051149_

Round 1

Reviewer 1 Report

The manuscript is very original and interesting, but the English should be revised by a native English speaker.   Three main remarks: i) in the abstract numeric data should be added as results; ii) the HPLC-MS methods should be improved by adding more information on validation and calibration parameters (or the authors should define if these methods were already validated in previous studies); iii) the conclusions should be written avoiding redundant information already discussed in the other sections and they should be focused only on the main results in relation to the aims of the study.   A last suggestion: Results and Discussion sections may be integrated into a single section to avoid repetitions.

Author Response

Point 1: in the abstract numeric data should be added as results;

Response 1: Thank you for your suggestion. We have added the numeric data in the abstract and marked up using the “Track Changes”.

Point 2: the HPLC-MS methods should be improved by adding more information on validation and calibration parameters (or the authors should define if these methods were already validated in previous studies);

Response 2: Thank you for your comment. We have added some parameters information to perfected the HPLC-MS methods in Materials and Methods section (Line 148-189).

Point 3: the conclusions should be written avoiding redundant information already discussed in the other sections and they should be focused only on the main results in relation to the aims of the study.

Response 3: Thank you for your comment. We have rewritten the conclusions to focus on the main results, and avoid repetitive information.

Point 4: Results and Discussion sections may be integrated into a single section to avoid repetitions.

Response 4: Thank you for your suggestion. We have carefully rewritten the Results and Discussion sections and deleted the duplicate parts. We read most of the articles in the journal and found that the results and discussion were independent, so the original format was still retained. We hope our modification can meet your requirements.

Reviewer 2 Report

Integrated Analysis of Gut Microbiome and Liver Metabolome to Evaluate the Effects of Fecal Microbiota Transplantation on Lipopolysaccharide/D-galactosamine-induced acute liver injury in mice

The manuscript was prepared very well. The introduction section justifies the purpose of the study. I congratulate the authors for the preparation of the manuscript

I would like to congratulate the authors for the structure of the manuscript and all the research carried out. It is highly publishable. However, there are some concerns, in part important, so the review articles need revision, see below.

Introduction

·       Why is this study considered relevant?

·       Indicate why this study is necessary.

·       I suggest that you incorporate a little more background related to the study.

Methods

·       Regarding the ethics committee for experiments, indicate the registration number

·       Indicate who handled the experimental animals and what accreditation they had

Results

·       The tables/figures and the text describing them do not require any input, it is the strongest part of this study. although it could more clearly describe the results

·       they are too small please enlarge them

·       You should improve the figure captions so that the reader understands the figures.

·       Add used abbreviations to the tables/figure

·       Discussion

·       Include a section on strengths / limitations.

·       Is it possible to describe more mechanisms responsible for the described actions?

·       What does this article contribute to, the authors should make their own assessment and include their own discussion of the results shown in the manuscript?

Conclusion

·        In the Conclusion section, state the most important outcome of your work. Do not simply summarize the points already made in the body — instead, interpret your findings at a higher level of abstraction. Show whether, or to what extent, you have succeeded in addressing the need stated in the Introduction (or objectives).

Author Response

Point 1: Introduction:

-Why is this study considered relevant?

-Indicate why this study is necessary.

-I suggest that you incorporate a little more background related to the study.

Response 1: Thank you for your comment. We have added more background in the introduction section, and marked up using the “Track Changes”. The effective therapeutic strategies for acute liver failure are still lacking new treatments and drugs are urgently needed. Liver disease and intestinal ecological disorders are strictly related, and the pathophysiology of this bidirectional relationship has received extensive attention. FMT can transfer intestinal microbes from a fit donor and possible implantation into the intestinal tract of recipients to rebuilt a healthy intestinal microbial ecosystem. And the function of FMT in the therapy of chronic hepatic disease and delaying the pro-gression of liver cancer also provides new horizons for the clinical therapy of ALF, as a viable, safe and low-cost treatment for a variety of liver diseases. In this study, we researched the preventive and therapeutic effects of FMT against LPS/D-gal caused ALF.

Point 2: Methods:

-Regarding the ethics committee for experiments, indicate the registration number

-Indicate who handled the experimental animals and what accreditation they had

Response 2: -ZJU20220438, and we have added the registration number in the paper (Line 84).

-Professors with Medical Laboratory Animal and animal Experiment Certificate of Zhejiang Province handled the experimental animals (Line 101-103).

Point 3: Results:

-The tables/figures and the text describing them do not require any input, it is the strongest part of this study. although it could more clearly describe the results

-they are too small please enlarge them

-You should improve the figure captions so that the reader understands the figures.

-Add used abbreviations to the tables/figure

Response 3: Thank you for your comment. We have enlarged the figures, and improved the figure captions, and added used abbreviations to the tables/figure.

Point 4: Discussion

-Include a section on strengths / limitations.

-Is it possible to describe more mechanisms responsible for the described actions?

-What does this article contribute to, the authors should make their own assessment and include their own discussion of the results shown in the manuscript?

Response 4: Thank you for your comment.

-We have added some strengths / limitations in Discussion (line 390-395).

-We have described more mechanisms in Discussion.

- We have explained some contributes (line 390-393).

Point 5: Conclusion

-In the Conclusion section, state the most important outcome of your work. Do not simply summarize the points already made in the body — instead, interpret your findings at a higher level of abstraction. Show whether, or to what extent, you have succeeded in addressing the need stated in the Introduction (or objectives).

Response 5: Thank you for your comment. We have rewritten the Conclusion section and deleted the duplicate points. Clarify the study of preventive and therapeutic effects of FMT against LPS/D-gal caused ALF, and the potential mechanisms of hepatoprotective effects of FMT.

Reviewer 3 Report

I reviewed the manuscript titled, Integrated Analysis of Gut Microbiome and Liver Metabolome to evaluate the effects of fecal microbiota transplantation on lipopolysaccharide/d-galactosamine-induced acute liver injury in mice. This study focused the integration of gut microbiome and liver metabolome to evaluate the effects of fecal microbiota transplantation on lipopolysaccharide/d-galactosamine-induced acute liver injury in mice. The study is appropriate and contributes to the field. In my opinion, this manuscript can be accepted for publication after addressing below comments/suggestions. All figures are extremely low quality and authors must improve the quality in order to recommend acceptance

Abstract:

Lines 27-30: should be revised to reflect the findings and provide recommendations

Introduction

Introduction is very short. Authors should focus on the background of the study, need of performing the research, research approach, research objectives. Please revise and add latest relevant studies  

Methodology

2.6. Microbial 16S rRNA Gene Sequencing Analysis… cite the reference

2.7. Metabonomics… cite the ref

2.8. Western blot analysis.. provide detailed methodology with ref

Results and discussion is appropriate

Figure 1: extremely low quality and authors must improve it

Figure 2: extremely low quality and not readable

Figure 3: extremely low quality and not readable

Figure 4. There is nothing to understand from the Figure. Very low quality. Without improvement in Figures, It is difficult to understand from the Figures.

Figures 5 and 6: The quality is extremely low. I recommend the editorial office to check the visibility of Figure before they send for peer-review. I cannot read the Figures even with 500% zoom with my clear eye vision. This is not the academic quality of presenting Figures in a manuscript.

Authors must improve the quality of all figures.

Conclusions should be revised to reflect the findings of the study

References are not according to the journal format.   

Author Response

Point 1: Abstract:

-Lines 27-30: should be revised to reflect the findings and provide recommendations

Response 1: Thank you for your comment. We have revised Lines 27-30 to emphasize the findings and recommendations, and marked up using the “Track Changes”.

Point 2: Introduction

-Introduction is very short. Authors should focus on the background of the study, need of performing the research, research approach, research objectives. Please revise and add latest relevant studies

Response 2: Thank you for your comment. We have added more background, need of performing the research, research approach and research objectives in the introduction section, and marked up using the “Track Changes”.

Point 3: Methodology

-2.6. Microbial 16S rRNA Gene Sequencing Analysis… cite the reference

-2.7. Metabonomics… cite the ref

-2.8. Western blot analysis.. provide detailed methodology with ref

Response 3: Thank you for your comment. We have added all the referencecs in the Methodology section.

Point 4: Results and discussion is appropriate

-Figure 1: extremely low quality and authors must improve it

-Figure 2: extremely low quality and not readable

-Figure 3: extremely low quality and not readable

-Figure 4. There is nothing to understand from the Figure. Very low quality. Without improvement in Figures, It is difficult to understand from the Figures.

-Figures 5 and 6: The quality is extremely low. I recommend the editorial office to check the visibility of Figure before they send for peer-review. I cannot read the Figures even with 500% zoom with my clear eye vision. This is not the academic quality of presenting Figures in a manuscript.

-Authors must improve the quality of all figures.

Response 4: We are very sorry for the low quality Figures1-5, and we have replaced all figures with higher quality ones in the papers.

Point 5: Conclusions should be revised to reflect the findings of the study

Response 5: Thank you for your comment. We have rewritten the conclusions to focus on the main results, and deleted repetitive information.

Point 6: References are not according to the journal format.

Response 6: We apologize for the wrong format of the reference, and we have modified it according to the format requirements of the journal.

Reviewer 4 Report

Good paper on a hot topic. I have some proposals.

Miinor points:

1. You discuss apoptosis. Please add a pra on te involvement of ferroptosis in the process.

2. Discussion. delete "etc" or add more information.

3. Add a para whether FMT is usefull in DILI:

doi: 10.3390/biomedicines11010015.  

Author Response

Point 1: You discuss apoptosis. Please add a pra on te involvement of ferroptosis in the process.

Response 1: Thank you for your comment. We are sorry that we have not found that FMT treatment of ALF is related to Ferroptosis before, so relevant indicators have not been determined. During the repair period, we do not have enough time to conduct experiments. If necessary, we will supplement experiments and further analyze.

Point 2: Discussion. delete "etc" or add more information.

Response 2: Thank you for your comment. We have delete "etc" in the discussion (line 508).

Point 3: Add a para whether FMT is usefull in DILI:  doi: 10.3390/biomedicines11010015.

Response 3: Thank you for your comment. We have discussed the effect of FMT on DILI (line 375-388).
